# Towards A Reconfigurable Systolic Array with Multi-Level Packing for Transformers

Tiandong Zhao*, Siyuan Miao*, Jialin Cao†, Shaoqiang Lu§, Jun Qiu‡, Xiao Shi‡, Kun Wang†, Lei He*§

*University of California, Los Angeles. *zhaotiandong@ucla.edu*
†Fudan University, China. *wangk@fudan.edu.cn*
‡Southeast University, China. *xshi@seu.edu.cn*
§Eastern Institute of Technology, China. *he@eias.ac.cn*

*Abstract*—Transformer-based models has achieved remarkable success in extensive tasks for natural language processing. To handle the variable-length sentences in human language, prior works suffer from low hardware efficiency due to either the shape mismatch between fixed-shape PEs (processing elements) and variable-shape workloads with data parallelism or large bubbles with pipeline parallelism. This ongoing work proposes a hybrid parallelism mixed with data parallelism for linear operators and pipeline parallelism for the attention. We develop a reconfigurable systolic array with multi-level packing to improve hardware efficiency. First, linear operators for different inputs can be packed along the array columns to improve spatial efficiency. Meanwhile, to boost temporal efficiency, we develop a head-level pipeline for attention with different stages packed on the array. We further skip the redundant computation in the masked attention by packing the computation of two heads along time. Packing decisions are explored with a dynamic programming based algorithm to maximize the overall throughput. Applied to GPT, our FPGA design has achieved $1.16\times$ higher normalized throughput and $1.94\times$ better runtime MAC utilization over the state-of-the-art GPU performance for variable-length sequences from MRPC, RTE and SQuADv2 datasets.

## I. INTRODUCTION

Transformer-based models have achieved remarkable triumphs in a wide range of deep learning tasks for natural language processing, such as machine translation [21], text classification [4] and generation [18], [19]. The extensive success is attributed to the task-agnostic model architecture with increasing number of encoder and decoder layers, and vocabulary size for better quality on various tasks. Such a trend, along with the unlimited text length that these language models need to handle, results in huge amounts of computation and parameters. The full GPT-3 [2] holds 175 billion parameters and requires $3.14 \times 10^{23}$ floating-point operations (FLOPS) for training, which would cost over \$4.6M using a Tesla V100 cloud instance for a single training run. The progressively higher computational demand calls for the need to exploit the efficiency of these models on devices.

The acceleration approaches in prior works fall into two paradigms. One approach exploits the intra-operator data parallelism (Fig.1a) in an operator-by-operator basis. [16] optimizes the operator partitioning for Transformer inference on TPUv4 [9]. [7], [10] boost GPU performance with tensor cores, while [3], [23] lay focus on the memory optimizations on GPU. [24]–[26], [29] develop specialized processing

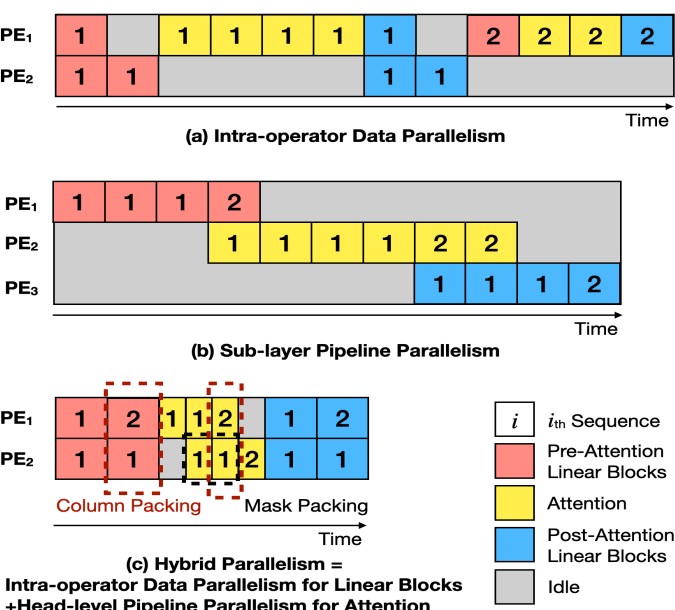

Fig. 1. Execution timeline for different parallel approaches. (a)Intra-operator data parallelism. (b)Sub-layer pipeline parallelism. (c) Hybrid parallelism. The time length of each block is only for illustration.

elements (PEs) for different operators. Padding arises from variable-length inputs in a batch due to the fact that popular deep learning frameworks [1], [14] can only handle rectangular shapes. The padded zeros thus introduce excessive overhead in both computation and memory. [6], [27] reduce the padding redundancy by reordering inputs during pre-processing, while [28] eliminates padding for linear blocks and fused attention on GPU by offsetting the variable-length inputs in memory. These works suffer from low efficiency either spatially from the mismatch between fixed-shape PEs and variable-shape workload, or temporally from non-overlapped memory access latency, especially in data-parallel fused attention.

Another approach resorts to the inter-operator pipeline parallelism (Fig.1b) where consecutive operators are assigned to different PEs. [5], [8], [12], [13] accelerate training of deep learning models with micro-batch layer pipeline. [15] constructs a sequence-wise sub-layer pipeline with approximated attention and feed-forward network. Since the computation of attention and linear blocks are at least linear complexity with

regard to the variable input length, pipeline parallelism at sub-layer level could inevitably result in severe pipeline bubbles for large length variance across input sequences.

We have two key observations on the GPU performance of Transformer-based models with intra-operator data parallelism. First, the attention suffers from both low temporal and spatial efficiency even for a fused kernel, indicating that the attention could potentially benefit more from inter-operator pipeline parallelism rather than intra-operator data parallelism. Second, the highly-optimized linear blocks only obtain 70% temporal efficiency and 25% spatial efficiency. Besides the shape mismatch, it is also limited by the capacity of shared memory and registers due to the fact that data parallelism leads to data replication, meaning less data reuse.

To address the above problems, we propose a hybrid parallelism (Fig.1c), data parallelism for linear blocks and pipeline parallelism for attention. The latter is finer-grained than sub-layer level to reduce pipeline bubbles. However, challenges arise in the architecture support for the hybrid parallelism. On one hand, inter-operator pipeline parallelism needs to *split* the PEs for pipeline stages. The shape mismatch can also be alleviated with the split along with workload decomposition. On the other hand, intra-operator data parallelism needs to *unify* the PEs and registers to maximize the data reuse. To meet the both requirements, we propose a runtime reconfigurable systolic array (RSA), where PEs across columns can work either together for a single operator or separately for multiple operators. Specifically, the RSA can be split for different input tokens in linear operators, or for different pipeline stages in the attention. We use *column packing* for this column-wise reconfigurable working pattern. Moreover, the masking in the decoder, which preserves the auto-regressive property to prevent leftward information in the flow, brings 50% redundancy in attention, especially for long sequences, but is neglected in prior works. We further propose *mask packing* to skip the redundant computation between two heads assigned to the same RSA columns.

Our contributions are summarized as follows:

- We develop a reconfigurable systolic array for hybrid parallelism, data parallelism for linear blocks and pipeline parallelism for attention, to improve the hardware efficiency of Transformer-based models.
- We propose a two-level packing, column packing and mask packing, to boost efficiency spatially and temporally for variable-length inputs. Packing decisions are explored with a dynamic programming based algorithm to maximize the overall throughput.
- Applied to GPT, our design on U200 FPGA shows $1.16\times$ higher normalized throughput and $1.94\times$ better runtime MAC utilization over the state-of-the-art GPU performance for variable-length input sequences from MRPC, RTE and SQuADv2 datasets.

In the following sections, we will first describe details of column packing and mask packing in Section II and then propose RSA architecture in Section III. Then we will explore the column packing decisions for hybrid parallelism in Section IV.Section V and Section VI present experiment results and conclusions.

## II. METHOD

### A. Column Packing

*1) Pack Linear Blocks:* We exploit the intra-operator data parallelism for each linear operator, namely a $M \times N \times K$ matrix multiplication(MM), where $M$,$N$,$K$ stand for input rows, output columns and hidden size. We have some observations on the MM shapes in a Transformer-based model. For a variable-length input, $M$ is equal to input sequence length $L$. Whether $N$ and $K$ are variable varies across different MMs. In the first case, which is also the most common case in the linear blocks of Transformer-based models, $N$ and $K$ are fixed as a multiple of head size $d_h$. The second case includes the two MMs in the attention with variable $N$ or $K$, where the shapes are $L \times L \times d_h$ and $L \times d_h \times L$. The shape mismatch between fixed-shape PEs and variable-shape MMs leads to low efficiency. Rather than suffering from multi-dimensional shape mismatch between PE and MM, we map the fixed shapes to the RSA rows and variable shapes to the RSA columns and temporal dimension so that the shape mismatch can be maximally alleviated by column packing. To be more specific, for a MM with variable-length inputs, we pack $N$ from different input sequences along RSA columns to maximize the spatial efficiency. We also take advantage of split-k, as described in [10], to partially unroll the $K$ dimension to balance the parallel workloads along columns for temporal efficiency.

*2) Pack Attention:* We propose a coarse-grained head-level pipeline for the attention with six stages, including $KQ$ load, MM $KQ^T$, $V$ load, softmax, MM $SV^T$ and final save. The two MMs are packed along RSA columns during the pipeline, where the former has variable $N$ and the latter has variable $K$. Since two different variable dimensions are mapped to RSA columns, weight stationary and output stationary dataflow are respectively required. Moreover, the number of heads to run per stage is worth study. More heads to pack in a MM stage leads to better spatial efficiency locally within the stage, but potentially results in worse global efficiency, since the larger pipeline granularity brings more bubbles. The pipeline stage partition will be discussed more in Section IV.

### B. Mask Packing

Each token only needs the computation results from its preceding tokens in the input sequence, but do not need those after. A Transformer decoder masks out the unnecessary ones to preserve the auto-regressive property. To eliminate the masking redundancy in $softmax(mask(KQ^T))V^T$, we propose mask packing as in Fig.2d. Rather than applying masking after the full computation of two $KQ^T$s, we skip the redundant computation and only generate a packed result matrix $S$. We use $S$ as packed layout for the following softmax and $SV^T$ for memory efficiency. So PEs need to handle $KQ^T$ and $SV^T$ with fixed and variable reduction length,

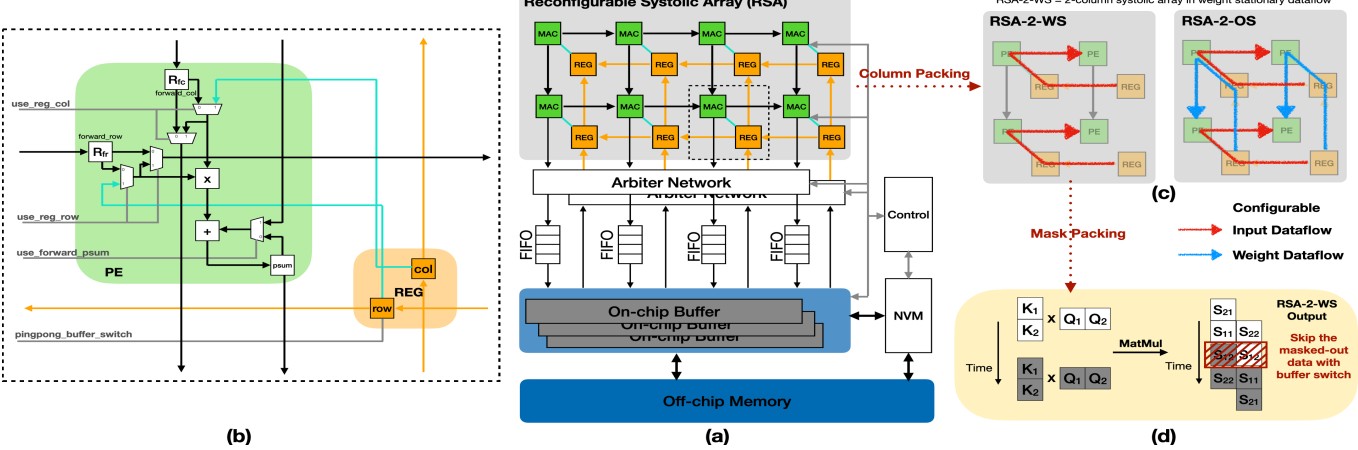

Fig. 2. (a) System diagram. (b) Circuit diagram of a RSA PE and its coupled shift registers. The input data path and buffer switch can be configured for packing. (c) RSA is split to halves in different dataflows for column packing. (d) Mask packing. We show an example where we skip the redundant computation for $KQ^T$ from two heads.

respectively, and the softmax module needs to handle vectors in the packed layout.

## III. HARDWARE ARCHITECTURE DESIGN

To provide the underlying architecture for the column packing and mask packing, we develop a RSA along with arbiter networks and a nonlinear vector module (NVM), as shown in Fig.2b.

We develop a two-dimensional systolic array with PEs and coupled shift registers. Fig.2b shows the circuit diagram of a RSA PE, which has three-level reconfigurablity from the control signals (gray). First, $use\_reg\_row$ configures the RSA split along columns by selecting the input data path to the multiplier. If it is set to 1, the multiplier will take the value stored in the row register (orange) as input via the reconfigurable data path (blue) rather than the forwarded value from the left PE, where the two neighboring PEs can thus work separate workloads. Second, the coupled shift register is for input buffering and its buffer switch can be configured for mask packing. Third, $use\_forward\_psum$ configures the dataflow for a RSA. If it is set to 1, PE uses the partial sum forwarded from upper PE and thus enables weight stationary dataflow. Otherwise, the accumulation will be performed locally as output stationary dataflow. The reconfigurable dataflow is for the two MMs in attention with variable $N$ and $K$ respectively so that they can be packed along RSA columns.

We use two arbiter networks for the interconnection between RSA and on-chip buffers to meet different communication patterns under column packing. For a MM where $K$ dimension is partially unrolled across columns, we need to collectively reduce the partial sums from multiple columns. For the attention pipeline, the result of the first MM computed on a RSA partition is written to on-chip buffers and then fed to another RSA partition. These two patterns are realized by the arbiter networks. Moreover, to handle the packed layout for mask packing, our NVM takes advantage of a configurable reduction

tree proposed in [17] for maximum and sum reduction with arbitrary length in softmax.

## IV. SCHEDULING

### A. Column Packing for A Single Operator

We first discuss the column packing decisions for a MM with variable-length inputs in the shape $L_i \times N \times K$. $L_i$ is the length of the $i$th input in a batch. Mapping $N$ and partial $K$ to the spatial column dimension and $L_i$ to the temporal dimension, we enumerate all combinations of $N$ values and $K$ factors to find the pair with maximal spatial efficiency. For example, we are mapping a MM with $N=4$ and $K=8$ to a RSA with 16 columns and 4 rows. To maximize spatial efficiency, we unroll $K=8$ along 2 columns besides 4 rows. Still, only 8 columns ($N=4 \times 2$, 2 is $K$'s column unroll factor) are used. So we split RSA to two partitions, each holding 8 columns and serving part of $L_i$ along the time. We then split the $L_i$ to two parts to balance the workload packed along columns.

### B. Column Packing for Attention Pipeline

For the attention pipeline at head level, we aim to split head sequences $\mathcal{H}=\{h_{ij}\}$, where $i$ is the sequence index in a batch and $j$ is the head index, to multiple stages, while minimizing the overall latency. Within each stage, intra-operator column packing in Section IV-A is applied. Mask packing is also applied to each MM. We formulate the pipeline stage partition as a dynamic programming problem. Its optimal sub-structure is listed in Eq.1. $p$ is a bit vector, where 1 means the $k$th head in $\mathcal{H}$ is packed with its last preceding head and 0 means no packing. The stage partition can be inferred from $p$ with simple union-and-find method. Column packing is constrained by the on-chip memory capacity. $M_{max}$ is the maximally allowed on-chip memory pressure and $M_k$ is the memory pressure of $k$th head. Iterating the $h_k$ in $\mathcal{H}$, we find the maximal overall throughput $\mathbb{T}$ with the head packed to last preceding head to one stage or not. If $h_k$ is packed, bookkeeping $p[k]=1$, $M_k$ is

on hold when exploring the column packing decision for next head. Otherwise, we check the packing of next head at a new stage with $M_{max}$. The optimal stage partition will maximally reduce the pipeline bubbles.

$$\mathbb{T}(k, p, M) = \max \begin{cases} \mathbb{T}(k-1, p[k] = 1, M - M_k) | M > M_k \\ \mathbb{T}(k-1, p[k] = 0, M_{max}) \end{cases}$$

(1)

## V. EXPERIMENT RESULTS

### A. Evaluation Setting

We implement our accelerator on Xilinx U200 FPGA with RSA in 4 rows and 1024 columns, NVM in 32 vector length, and four DDR4. The latency in cycle count for the evaluation below is collected through RTL-level simulation with Xilinx Vivado Suite.

We benchmark our design with 6 different settings on input sequence length. The first three are constructed with fixed-length inputs in length 64, 512, 2048 in batch size 8. The other three collect variable-length inputs from MRPC, RTE and SQuADv2 [20] test sets, respectively, and then packed in batches with size 8. Three datasets have 14/40, 54/240, 167/791 for average/maximal sequence length. The former two datasets are from GLUE [22] benchmark suite with representative length in small and medium length, while the latter covers more long length. We run these datasets on a small GPT-3 for evaluation. The model includes 12 layers, 768 embedding size, 12 heads and 64 head size.

### B. Performance with Step-wise Optimization

We apply step-wise evaluation to show the effect of hybrid parallelism with column packing and mask packing. *Intra-operator Data Parallel* runs the model on RSA with intra-operator data parallelism in operator-by-operator basis. *Layer Pipeline* runs a two-stage layer pipeline parallelism, where the RSA is split to halves and each runs a Transformer layer for different sequences. The other three applies hybrid parallelism incrementally with different packing methods.

Fig. 3 shows the impact of step-wise optimization on GPT. One can find that *Layer Pipeline* is limited by off-chip memory bandwidth on a single device and thus performs worse for longer sequences. The hybrid parallelism is effective in all cases with $1.17\times$ higher throughput on average than intra-operator data parallelism, while column packing and mask

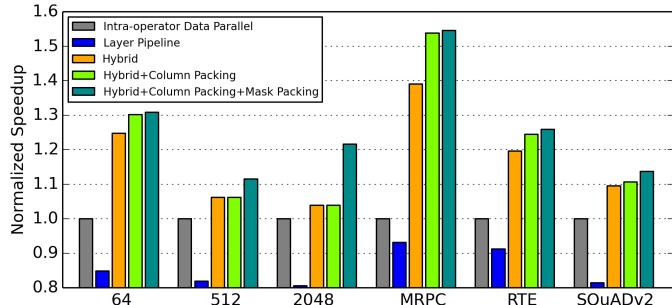

Fig. 3. GPT performance on RSA with step-wise optimization.

packing bring $1.21\times$ and $1.26\times$ performance boost respectively. Column packing gets more benefits for short sequences, such as fixed 64, MRPC and RTE, with better spatial efficiency. Mask packing benefits more for long sequences, which brings additional 30% for fixed 2048, but is marginal for other cases. This is because the computation of attention grows quadratically with sequence length and the attention takes 50% of total computation for fixed 2048, while it takes less than 10% for fixed 64.

### C. End-to-End Performance

Table I compares our performance with other works on GPU and FPGA on GPT. Batch size 8 is used for all cases. We evaluate GPU performance with [28], which is the state-of-the-art GPU work for variable-length inputs. [15] is optimized for variable-length inputs on FPGA, but does not include detailed throughput for each dataset. [11] only reports performance for fixed-length inputs, which is also the case for other FPGA works. So we compare the performance for fixed 128 length inputs with [11] and that for variable-length inputs with [28] on three datasets. The throughput is normalized in MAC units and 16-bit precision. Our design outperforms GPU and FPGA works by $1.16\times$ and $2.11\times$ on normalized throughput, respectively, across fixed-length and variable-length inputs. The advantage comes from the better efficiency due to the column packing and mask packing on our RSA, which shows $1.94\times$ and $1.18\times$ better MAC efficiency over GPU and FPGA works. This observation exhibits the advantage of our RSA architecture over others for Transformer-based models with variable-length inputs.

| Input Sequence | Fixed 128 | | | MRPC | | RTE | | SQuADv2 | |
|---|---|---|---|---|---|---|---|---|---|
| Platform | [28] | [11] | Ours | [28] | Ours | [28] | Ours | [28] | Ours |
| Device | A100 GPU | ZCU102 FPGA | U200 FPGA | A100 GPU | U200 FPGA | A100 GPU | U200 FPGA | A100 GPU | U200 FPGA |
| Precision | FP16 | INT8 | INT16 | FP16 | INT16 | FP16 | INT16 | FP16 | INT16 |
| Frequency(MHz) | 1095 | 214 | 200 | 1095 | 200 | 1095 | 200 | 1095 | 200 |
| Tensor Core/DSP | 432 | 3287 | 4160 | 432 | 4160 | 432 | 4160 | 432 | 4160 |
| Runtime Utilization (FLOPS/MAC) | 0.60 | 0.79 | 0.93 | 0.21 | 0.66 | 0.34 | 0.70 | 0.54 | 0.75 |
| Normalized Throughput (GFLOPS/MAC) | 0.16 | 0.09 | 0.19 | 0.11 | 0.13 | 0.10 | 0.14 | 0.14 | 0.15 |

TABLE I
COMPARE END-TO-END PERFORMANCE WITH GPU AND OTHER FPGA WORKS.

## VI. CONCLUSION

We propose a hybrid parallelism for Transformer-based models with variable-length inputs, specifically data parallelism for linear operators and pipeline parallelism for attention. To make it happen, we develop a reconfigurable systolic array with multi-level packing. First, for a single linear operator, we pack the computation of different input sequences along the array columns for spatial efficiency. Second, to improve the temporal efficiency of the attention block, we develop a head-level pipeline with stages packed along the array columns. Moreover, we develop mask packing to skip the redundant computation that are masked out by Transformer decoder masking. Column packing decisions are explored with a dynamic programming based algorithm to maximize the overall throughput. Applied to GPT, our design on Xilinx U200 FPGA outperforms state-of-the-art GPU work for variable-length inputs by $1.16\times$ in normalized throughput and $1.94\times$ in runtime MAC utilization across MRPC, RTE and SQuADv2 datasets.

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
