# OpenReview forum: "Towards A Reconfigurable Systolic Array with Multi-Level Packing for Transformers"
_iscaconf.org/ISCA/2023/Workshop/ASSYST — ASSYST Oral_

### Official Review · Reviewer_RYNk · 2023-05-05
**Interesting paper but clarity needs some improvement**

**Rating:** 6
**Confidence:** 3

**Review:**

I liked this paper. The problem is well-motivated and the evaluation is thorough, with FPGA synthesis and detailed simulation along with sensitivity studies to show the impact of various optimizations. The authors demonstrate a reasonable improvement relative to an A100 GPU.

I found some parts of this work slightly hard to understand. In particular, the dynamic programming for packing stages was unclear, and I think it could benefit from a diagram and possibly a more compelling argument for/proof of optimality. I was also unsure how IV.A's "enumerate all combinations" scales in terms of complexity and evaluation time.

A few hardware-related questions:
1. Is the RSA configured globally or per-element, and what impact (if any) does the reconfiguration time have on the results?
2. How would the packing algorithm's efficiency change with different layouts? 4x1024 is a very wide layout, and if the RSA is being partitioned on a column basis this would result in a very fine-grained partitioning. More square arrays (e.g., 8x512 or 16x256) may not show the same efficiency gains when partitioning on a column basis.
3. What's the size of the arbiter network? If this is a 1024x1024 network, it could be providing a significant advantage.
4. Why simulate instead of running the experiments on a real FPGA---was the design placed and routed or only synthesized? What is the area breakdown of components?

**Review (Strengths/Weaknesses):**

# Strengths
- Speedup relative to A100 is impressive
- Well-motivated experiments and evaluation
# Weaknesses
- I wish there were an area breakdown of the accelerator logic on the FPGA and sensitivity studies for various hardware sizes/configurations (rows/columns).
- The A100 results are showing FP16 performance while the FPGA results are showing INT16 performance
- Writing is unclear, especially for the dynamic programming to schedule the dataflow computation
- A few questions remain about evaluation generality

**Reviewer Expertise:**

Knowledgeable: I used to work in this area and/or I try to keep up with the literature but might not know the latest developments.

---

### Official Review · Reviewer_eC61 · 2023-05-05
**Review for Towards A Reconfigurable Systolic Array with Multi-Level Packing for Transformers**

**Rating:** 7
**Confidence:** 3

**Review:**

The paper proposes a mixed execution and data parallelism approach combined with a dynamic data packing algorithm to accelerate Transformer models. They evaluate their scheme on a runtime-reconfigurable systolic array fabric, implemented on an FPGA, and demonstrate speedups over comparable GPU and FPGA implementations.



**Review (Strengths/Weaknesses):**

### Strengths
+ The hardware implementation is evaluated on an FPGA, with numbers reported from RTL simulation
+ The evaluation is clear and includes detailed ablation results of the contributions

### Weaknesses
- Since the paper is described as being work-in-progress, the paper could benefit from a more detailed discussion of the limitations and future directions of the proposed approach.

### Questions
- Why was int16 chosen as the bitwidth for the accelerator?
- Were there any design decisions that were constrained by the specific FPGA used for implementation? For this design, would the Xilinx U200 be sufficient for models larger than GPT-3?

**Reviewer Expertise:**

Knowledgeable: I used to work in this area and/or I try to keep up with the literature but might not know the latest developments.

---

### Official Review · Reviewer_3xRw · 2023-05-06
**Towards A Reconfigurable Systolic Array with Multi-Level Packing for Transformers**

**Rating:** 5
**Confidence:** 3

**Review:**

This paper presents hardware optimizations for transformer-based large language models on FPGAs, focusing on spatial and temporal utilization. The key idea is presented to be the reconfigurable systolic array design which allows parallelism in both column and row order where the PEs can be configured to operate in parallel or in a pipeline. PE utilization is considered for variable length inputs, self attention partitioning and masking. Fusing masking into matrix multiply saves redundant computations. A dynamic programming algorithm is used to find efficient packing decisions for the PEs.

**Review (Strengths/Weaknesses):**

Proposed optimizations sound viable. The results show significant improvement over the baseline. However, all of the presented numbers are normalized to their own baseline. And comparisons are performed in terms of MAC/FLOPS utilization of the individual platform (GPU or FPGA). There is no single absolute latency or throughput number presented. Which makes the evaluation weaker and harder to understand where the optimizations stand compared to SOTA.


**Reviewer Expertise:**

Knowledgeable: I used to work in this area and/or I try to keep up with the literature but might not know the latest developments.